# Acceptability of aspirin for cancer preventive therapy: a survey and qualitative study exploring the views of the UK general population

Kelly E Lloyd [iD],[1] Louise H Hall,[1] Lucy Ziegler,[1] Robbie Foy [iD],[1] Sophie M C Green,[1] Mairead MacKenzie,[2] David G Taylor,[3] Samuel G Smith [iD],[1] Aspirin for Cancer Prevention AsCaP Steering Committee

[1]Leeds Institute of Health Sciences, University of Leeds Faculty of Medicine and Health, Leeds, UK
[2]Independent Cancer Patients' Voice, London, UK
[3]School of Pharmacy, University College London, London, UK

**Correspondence to**
Dr Kelly E Lloyd;
k.e.lloyd@leeds.ac.uk

## ABSTRACT

**Objectives** Aspirin could be offered for colorectal cancer prevention for the UK general population. To ensure the views of the general population are considered in future guidance, we explored public perceptions of aspirin for preventive therapy.

**Design** We conducted an online survey to investigate aspirin use, and awareness of aspirin for cancer prevention among the UK general population. We conducted semistructured interviews with a subsample of survey respondents to explore participants' acceptability towards aspirin for cancer preventive therapy. We analysed the interview data using reflexive thematic analysis and mapped the themes onto the Theoretical Domains Framework, and the Necessity and Concerns Framework.

**Setting** Online survey and remote interviews.

**Participants** We recruited 400 UK respondents aged 50–70 years through a market research company to the survey. We purposefully sampled, recruited and interviewed 20 survey respondents.

**Results** In the survey, 19.0% (76/400) of respondents were aware that aspirin can be used to prevent cancer. Among those who had previously taken aspirin, 1.9% (4/216) had taken it for cancer prevention. The interviews generated three themes: (1) perceived necessity of aspirin; (2) concerns about side effects; and (3) preferred information sources. Participants with a personal or family history of cancer were more likely to perceive aspirin as necessary for cancer prevention. Concerns about taking aspirin at higher doses and its side effects, such as gastrointestinal bleeding, were common. Many described wanting guidance and advice on aspirin to be communicated from sources perceived as trustworthy, such as healthcare professionals.

**Conclusions** Among the general population, those with a personal or family history of cancer may be more receptive towards taking aspirin for preventive therapy. Future policies and campaigns recommending aspirin may be of particular interest to these groups. Multiple considerations about the benefits and risks of aspirin highlight the need to support informed decisions on the medication.

## STRENGTHS AND LIMITATIONS OF THIS STUDY

⇒ The online survey collected unique data on aspirin use and reasons for taking aspirin among the UK general population aged 50–70 years.

⇒ The interviews offered the opportunity to explore the potential barriers and facilitators to taking aspirin for cancer prevention among the general public.

⇒ Self-selection to the survey and interviews may have resulted in recruiting participants with stronger views on aspirin than the wider UK population.

## INTRODUCTION

Colorectal cancer is one of the most common cancers worldwide, with an estimated two million cases and nearly one million deaths from the disease globally in 2020.[1] There is increasing interest in the pharmacological prevention of cancer,[2] including aspirin to prevent colorectal cancer.[3] A pooled analysis of 423 495 people from 2 cohort studies found that daily aspirin use was associated with a 15% reduced risk of colorectal cancer (HR: 0.85, 95% CI=0.80 to 0.89).[4] Furthermore, a meta-analysis of four randomised controlled trials investigating aspirin for vascular disease prevention found that aspirin was associated with reduced colon cancer incidence (HR: 0.76, 95% CI=0.60 to 0.96).[5] Studies have investigated the use of aspirin for preventing other cancers; however, the evidence is more limited.[6]

Aspirin is often recommended for people at higher risk of developing colorectal cancer. Australian national guidance[7] and the UK National Institute for Health and Care Excellence (guidance NG151) recommend aspirin for colorectal cancer prevention among people with Lynch syndrome.[8] The guidance does not state a recommended dose, but 150–300 mg is commonly used in

practice. Australian guidance also recommends considering 100–300 mg daily aspirin for those in the general population aged 50–70 years to prevent colorectal cancer.[7] There is no current equivalent national guidance for the public in the UK.

Decisions on whether to use aspirin involve consideration of potential benefits and side effects, such as an increased risk of gastrointestinal bleeding.[9] For individuals at population risk of colorectal cancer, regular aspirin use between 75 and 325 mg appears to have a favourable benefit–harm profile,[6] although the risk of side effects increases substantially after age 70.[6 10 11] The current use and acceptability of aspirin for colorectal cancer prevention among the UK public is unknown.[12] It is likely though that a proportion of the public have experience taking aspirin,[13] such as for pain relief, or cardiovascular disease risk reduction.[14] People's prior use of aspirin could support the implementation of the medication for the purpose of colorectal cancer prevention.

Most research examining barriers to using preventive therapy has focused on pharmacological prevention of breast cancer among women at higher risk of the disease. Barriers to use include concerns about side effects,[15–18] and perceptions of the medication as a 'cancer drug'.[15] Research has also explored the views of people with Lynch syndrome on aspirin for colorectal cancer prevention.[19] Several barriers to taking aspirin were observed, including concerns regarding aspirin's side effects and confusion on the recommended dose. It is important to explore the views of the general population on aspirin for cancer prevention, as there are likely to be similar and different barriers to initiating aspirin compared with those at higher risk. Australian cross-sectional research has observed moderately high acceptance (>70%) for taking aspirin regularly for colorectal cancer prevention among the general population.[20] However, the study did not explore participants' motivators and barriers towards the use of aspirin.

The potential impact of using aspirin for cancer prevention in the wider population will depend on acceptability as well as effectiveness, and an understanding of the barriers to implementation. Public perceptions of aspirin for cancer preventive therapy should be explored to inform both clinical guideline development[21] and support informed decision-making.[22] In this study, we aimed to investigate aspirin use and associated knowledge among people from the UK general population. We also aimed to explore the potential facilitators and barriers towards taking aspirin for colorectal cancer prevention.

## METHODS
### Design
We carried out a survey and qualitative study. We conducted an online survey for two reasons: (1) to collect initial data on participants' prior use of aspirin and their knowledge on taking aspirin for cancer prevention and (2) to recruit people from the UK general population to an interview.

Following the survey, we conducted theory-informed, semistructured, one-to-one interviews with a subsample of survey participants to explore their perceptions of aspirin for colorectal cancer prevention. We preregistered the methods and analysis plan (https://doi.org/10.17605/OSF.IO/3EFG7). We initially preregistered the study to explore the views of people with Lynch syndrome, members of the public and healthcare professionals on aspirin. We deviated from the protocol by publishing the Lynch syndrome and healthcare professional findings in a separate paper.[19] Study reporting was guided by the Standards for Reporting Qualitative Research checklist.[23]

### Participants and recruitment
We hosted the online survey on Qualtrics, and recruitment was advertised through a market research company (Dynata). We recruited people from the UK public between the ages of 50 and 70 years, as the benefits of prophylactic aspirin use are estimated to be greater than the risks of side effects within this age range.[6] At the end of the survey, we asked respondents for contact details if they wished to take part in a follow-up interview. All interviewees received £25 from Dynata. One author (KEL) recruited participants until data saturation was considered to have been achieved, following an established method for assessing data saturation.[24] Data saturation was used to ensure that the sample size was large enough to provide informative data on the topic, but not so large as to waste resources and participants' time. We used a 10+3 stopping criterion to assess for data saturation, which is a tested and recommended approach for theory-based interview studies.[24] After 10 interviews, we assessed for data saturation and stopped recruitment once three subsequent interviews had been conducted and no new themes were identified.[24] For example, recruitment would stop after participant 13, if no new themes were observed from participants 11, 12 and 13.

### Survey measures
#### Aspirin use
We asked participants whether they had ever taken aspirin. Those who answered yes were asked if they took aspirin regularly (ie, most days or every day) and their reasons for taking aspirin, such as for pain relief, cancer prevention or cardiovascular disease prevention (online supplemental materials 1).

#### Knowledge
We asked participants if they were aware prior to the survey that aspirin can be used to reduce the risk of developing certain cancers. We also asked whether a healthcare provider had previously discussed with them about taking aspirin to prevent colorectal and other cancers.

#### Characteristics of the sample
We collected data on participant characteristics, including any previous cancer diagnoses, gender, age, ethnicity and highest educational or professional qualification obtained.

## Interview schedule

We developed the interview schedule based on the Theoretical Domains Framework (TDF; V.2),[25] which is a framework derived from multiple behaviour change theories (online supplemental materials 2). Each of the 14 framework domains describes a factor that could influence individual behaviour when implementing new clinical practices. The TDF domains cover both internal factors, such as a person's knowledge and emotions, and external factors, such as social influences and available resources.

We conducted semistructured interviews with flexibility to the order of questions, and improvised follow-up questions. At the beginning of all interviews, participants were informed that aspirin is currently only recommended in the UK for people at higher risk of colorectal cancer due to a genetic syndrome, but that there is potential for wider recommended use in the future.

## Patient and public involvement

A public representative (MM) reviewed the draft interview schedule to ensure the questions were comprehensible to participants.

## Data collection and analysis

One author (KEL) analysed the survey data in R Studio (R V.4.2.1), with findings presented in proportions and frequencies. One author (KEL), with previous experience in collecting and analysing qualitative data, conducted all interviews by video call or telephone. Interviews took place from July to August 2022. Interviews were audio recorded, transcribed verbatim and pseudonymised using initials to replace participants' names.

The interview data was analysed in two stages, which is recommended for the TDF to optimise its use in qualitative research[26] and can identify themes not captured by a theoretical framework.[26 27] The interview transcripts were coded and analysed using reflexive thematic analysis.[28 29] The developed themes were then mapped onto the TDF domains.[25] During analysis, we found the Necessity–Concerns Framework to be an additional useful framework for guiding our analytic process.[30] The framework specifies that people consider their treatment necessity beliefs against their concerns when making decisions on medication[31 32] and that these beliefs and concerns on the medication can be general or specific. One author (KEL) coded and analysed all transcripts, and two authors (SGS and SMCG) double coded a proportion of interviews. The findings and potential themes were discussed collaboratively among the three authors, with the aim to explore new perspectives on the data and enhance interpretative depth.[33] One author (KEL) mapped the themes onto the domains in the TDF (V.2), which was reviewed by coauthors. All transcripts were managed in NVivo (V.1.6.1) and Microsoft Word.

The survey dataset and analysis scripts generated for the study are publicly available in the Research Data Leeds Repository.[34]

**Table 1** Characteristics of the survey respondents, recruited from the UK general population (n=400)

| | n (%) |
|---|---|
| **Age** | |
| 50–55 | 88 (22.0%) |
| 56–60 | 106 (26.5%) |
| 61–65 | 103 (25.8%) |
| 66–70 | 103 (25.8%) |
| **Gender** | |
| Women | 186 (46.5%) |
| Men | 213 (53.3%) |
| Another identity | 1 (0.3%) |
| **Ethnicity** | |
| White British or Irish | 371 (92.8%) |
| White Gypsy or Irish Traveller | 1 (0.3%) |
| Any other white background | 12 (3.0%) |
| Asian or Asian British | 9 (2.3%) |
| Mixed white and Asian | 1 (0.3%) |
| Arab or Arab British | 2 (0.5%) |
| Black/African/Caribbean background/ mixed background | 2 (0.5%) |
| Any other ethnic background/mixed background | 2 (0.5%) |
| **Education** | |
| Degree level and above | 149 (37.3%) |
| Below degree level | 251 (62.8%) |
| **Previously diagnosed with cancer** | |
| Yes | 46 (11.5%) |
| No | 354 (88.5%) |
| **Among those 'yes' to cancer (n=46), which cancer(s)** | |
| Prostate | 11 (23.9%) |
| Breast | 10 (21.7%) |
| Skin | 6 (13.0%) |
| Colorectal | 5 (10.9%) |
| Other cancers | 14 (30.4%) |

Proportions may not compute to 100% due to rounding.

## RESULTS

### Survey findings

In total, 400 people participated in the survey (table 1). The mean age of the sample was 60.8 years (SD=5.7). Most of the sample were men (213; 53.3%), educated below degree level (251; 62.8%) and were white British or Irish (371; 92.8%). In total, 76 (19.0%) were aware prior to the survey that aspirin can be used to reduce the risk of developing certain cancers and 15 (3.8%) had previously discussed aspirin for this purpose with a healthcare professional.

Most participants (216; 54.0%) had taken aspirin at least once. Among the 216 previous aspirin users, most

**Table 2** Characteristics of the general public interview respondents (n=20)

|  | n (%) |
|---|---|
| **Age** |  |
| 50–55 | 3 (15.0%) |
| 56–60 | 7 (35.0%) |
| 61–65 | 5 (25.0%) |
| 66–70 | 5 (25.0%) |
| **Gender** |  |
| Women | 12 (60.0%) |
| Men | 8 (40.0%) |
| **Ethnicity** |  |
| White British or Irish | 17 (85.0%) |
| Asian or Asian British | 3 (15.0%) |
| **Country in the UK** |  |
| England | 18 (90.0%) |
| Scotland | 1 (5.0%) |
| Northern Ireland | 1 (5.0%) |
| **Previously diagnosed with cancer** |  |
| Yes | 6 (30.0%) |
| No | 14 (70.0%) |

had taken it for pain relief (150; 69.4%). Fewer had taken aspirin for prevention of cardiovascular disease (61; 28.2%) and cancer (4; 1.9%). One person (0.5%) had taken aspirin as part of a trial. Among the 216 previous aspirin users, 59 (27.3%) reported using aspirin regularly, defined as most days or every day. Most of the 59 regular aspirin users took it for cardiovascular disease prevention (45; 76.3%). Other reasons for regularly taking aspirin were pain relief (11/59, 18.6%) and cancer prevention (3/59; 5.1%).

### Interview findings

A total of 202 (50.5%) survey respondents expressed interest in a follow-up interview. We invited participants by email to be interviewed until we judged data saturation was reached. We purposefully sampled participants from different demographic groups (eg, gender) and a balance of current and never aspirin users. In batches of 5–10 invitations, we invited 53 survey respondents to be interviewed and 20 (37.7%) responded and were interviewed (table 2). Interview duration ranged between 15 and 30 min. We identified three overarching themes, and interview findings were mapped onto the TDF (table 3).

#### Perceived necessity of aspirin
##### Did not perceive a need for aspirin

Participant beliefs varied about the perceived necessity of taking aspirin for colorectal cancer prevention. Several people perceived themselves to be at low risk of the disease, typically because they had no family history or personal history of cancer. Often, these participants described how they would only consider using aspirin for cancer prevention if a medical doctor assessed them to be at higher risk, or if they were diagnosed with colorectal cancer.

> If someone said to me I was high risk and aspirin was the, was a possible option, I would take it. (J.J., 56–60 years).

> Yeah, well I think probably if I have had bowel cancer and I was in recovery […] then I probably would take [aspirin]." (W.G., 66–70 years).

Several participants were averse to using daily medication in general, unless deemed highly necessary for their health.

> I would only take [medication] if it was really necessary. (A.G., 66–70 years).

**Table 3** The themes, and corresponding barriers, facilitators and domains within the Theoretical Domains Framework (TDF; V.2)

| Themes | Potential barriers to the use of aspirin for preventive therapy | Potential facilitators to the use of aspirin for preventive therapy | Main TDF domain(s) |
|---|---|---|---|
| Perceived necessity of aspirin | Those who perceive themselves at lower risk of cancer because they do not have a personal or family history of the disease. | Those who perceive themselves at higher risk of cancer because they have a personal or family history of the disease. | Beliefs about consequences. |
| Concerns about side effects | High concerns about using daily aspirin because of the side effects, such as increased risk of gastrointestinal bleeding. Concerns about taking aspirin daily at doses above 75 mg. | Low concerns regarding the side effects of aspirin because it is a well-known over-the-counter medication for pain relief. | Beliefs about consequences. |
| Preferred information sources | Current lack of a recommendation to support the use of aspirin for colorectal cancer prevention among the UK general public. | Wanting information on aspirin to come from trusted online sources of information, such as the National Health Service and UK cancer charities (eg, Cancer Research UK). Wanting information on aspirin to come from a medical professional. | Environmental context and resources. |

Table adapted from Burgess *et al*.[60]

On the whole I don't like taking pills, […]. Unless there was very definite proof that not taking [aspirin] would lead to bowel cancer. (W.G., 66–70 years).

A few participants perceived other lifestyle changes, such as diet and attending preventive screening, as more effective and necessary than taking medication to prevent cancer.

I think [colorectal cancer] can be broadly governed by diet, fibre, etc. (E.Y., 56–60 years).

So I'm at the age now where I get a bowel screening, […] So I'm having difficulty seeing what aspirin would add to the mix if I'm kind of okay at the moment. (R.C., 56–60 years).

### Perceived a necessity for aspirin

A number of participants discussed having previously been diagnosed with cancer (eg, breast, colorectal, prostate), which in turn increased their interest in using aspirin, and their feelings on the importance of cancer prevention.

If I saw the GP yesterday […] 'oh would you like to take aspirin as a preventative for bowel cancer?' I would say yes, particularly now that I've been diagnosed with, with breast cancer. (I.N., 61–65 years).

Several participants appeared more inclined to use aspirin because friends or family members had been diagnosed with cancer. All three interviewees who currently used or had previously used aspirin for cancer prevention had started because of a family history of cancer.

A few years ago I heard an article on [a public service radio station] and there was two eminent doctors, and they were talking about the possibility that aspirin prevented not just bowel cancer but other cancers. […] That triggered me to start taking it, my father died of cancer and my sister also died of cancer. (H.B., 66–70 years).

### Concerns about side effects
### Expressed high concerns about aspirin

Many participants expressed concerns about the potential side effects of daily aspirin use. Several participants expressed feeling more comfortable using a lower dose of daily aspirin (eg, 75 mg), due to the concerns about these side effects.

Because [150-300mg] seems to be a lot. […], but 75mg, a small dose, I would feel comfortable with that. (S.S., 66–70 years).

Several participants' concerns regarding aspirin appeared to be related to the perceived harm of taking any long-term medication, and the perception that taking daily medication would mean that they were an 'unhealthy' person.

[From taking medication] you suddenly step from being a healthy person to a potentially unhealthy

person because it's a medical intervention (R.C., 56–60 years).

Most participants wanted further information on the side effects of aspirin before they would consider initiating the medication. Participants often wanted to know if the benefits of aspirin for preventive therapy substantially outweighed the side effects.

I probably would like to know more about what effects it would have on my body, what damage it could do to my heart, or other organs. (K.H., 56–60 years).

### Did not have concerns about aspirin

Despite widespread concerns about aspirin, these views were not universally held among participants. Aspirin was a familiar drug to many, which lowered some concerns about side effects.

Because it's been well known to me that [aspirin] can help prevent heart problems and that's kind of been around for decades […] So I think knowing that and not hearing of anything happening to anybody then that's very reassuring for me." (R.L., 56–60 years).

Some participants expressed low concerns about aspirin because of their own or family members previous experiences using the medication without encountering side effects.

No, no I don't [have concerns about the side-effects], because you know whenever I've taken it, […] I don't take them every day but my mother had no side-effects whatsoever. (I.N., 61–65 years).

### Preferred information sources

Participants discussed their preferred source of information on the use of aspirin for colorectal cancer prevention. Many participants wanted the information on aspirin to come from trusted online sources, such as the National Health Service and UK cancer charities (eg, Cancer Research UK).

Cancer Research would probably be, or Macmillan, it would be a trusted website that […] I would look at. (K.H., 56–60 years).

Other important information sources discussed were medical professionals. Several participants wanted the information on aspirin for cancer prevention to come from their general practitioner (GP), while others preferred speaking to a cancer specialist. In most cases, participants felt that they would not initiate aspirin for preventive therapy without a doctor's recommendation.

Well certainly I think I would first and primarily go to the GP. (A.N., 66–70 years).

Maybe not a GP but somebody who was specialised in that area, maybe a gastroenterologist […] I wouldn't automatically [take aspirin] because I wouldn't have

enough information to warrant what else it could do to me. (K.H., 56–60 years).

Several people expressed positive views towards speaking to a pharmacist about aspirin, often because they had previously had positive experiences consulting their pharmacists on other medical issues. However, not all were comfortable with this approach, and only wanted to speak with a medical doctor about aspirin.

Well I think my doctor is the trustworthy cause, but also my pharmacy is really good. (A.G., 66–70 years).

I've always gone to GP, I've never discussed anything with a pharmacist ever […] I mean they're giving a lot of powers to the pharmacist, but I don't feel comfortable talking to a pharmacist. (L.L., 61–65 years).

Official UK guidance recommending aspirin for cancer prevention for the general public was an important factor to some participants. However, this was only mentioned by a small number of people as a potential barrier to using aspirin.

I'd be happy if [aspirin] was recommended though, but I'd wait till it was recommended, I wouldn't jump on things too quickly. (F.L., 50–55 years).

## DISCUSSION

This study of the UK general population found low reported current use of aspirin for cancer prevention among survey respondents, and several barriers and motivators towards taking aspirin for this purpose across interviewees. In the interviews, those with a personal or family history of cancer were more receptive to taking aspirin while those who considered themselves at lower risk of colorectal cancer, or cancer in general, expressed more resistance. Public perceptions towards taking aspirin for cancer prevention should be considered in any future guidance recommending aspirin outside of a Lynch syndrome population.

Most survey respondents had taken aspirin, often for pain relief. People's perceptions of aspirin as a well-known medicine may support its use for prevention, as several participants interviewed held positive views towards aspirin for this reason. However, many had concerns about side effects, highlighting the need to support informed choices about taking medication. Awareness of aspirin for cancer prevention among survey respondents was low, and only a small percentage took aspirin for prevention. The interview findings suggest family history of cancer was an important motivator to taking aspirin among those at population risk.

We considered our qualitative findings in relation to the 14 domains in the TDF, and identified 2 main domains.[25] These domains were the 'beliefs about the consequences' of using aspirin and the potential 'environmental context and resources' which would aid in implementing aspirin for the public. Participant beliefs about the consequences

of taking aspirin, such as the necessity for and side effects of the medication, were particularly important and relate to the Necessity–Concerns Framework.[30] Several participants expressed specific beliefs and concerns on taking aspirin for cancer prevention, as well as general concerns on taking any medication daily. Previous evidence has found those who report low necessity for a medication and high concerns on the side effects are less likely to initiate and adhere to a range of different medications.[35–38] Similarly, a UK prospective study found women at higher risk of breast cancer with low concerns about side effects were significantly more likely to initiate preventive medication.[39]

While the interview findings suggest a relationship between perceived cancer risk and uptake of aspirin, a large population survey is warranted to investigate these potentially motivating factors further. Although minimal previous research has been conducted in this area,[12] surveys in the USA and Australia have observed mixed findings on a relationship between cancer risk and interest in aspirin for colorectal cancer prevention.[20 40] Interventions aiming to support informed uptake of aspirin for preventive therapy should consider targeting people's beliefs regarding the side effects and benefits of taking the medication. There is also scope for research to explore the relationship between the Necessity–Concerns Framework and adherence to aspirin.

Guidance recommending aspirin for preventive therapy among the public should consider how the information is communicated, as several participants want the advice to come from a healthcare professional. People who have previously had cancer or a precursor to cancer may be particularly receptive to taking aspirin for preventive therapy. There is trial and observational evidence supporting the use of aspirin among people with colorectal adenomas,[41–47] and this may be an appropriate group for policy-makers to target if the harm–benefit profile is deemed sufficient. While the evidence for using aspirin for secondary cancer prevention is less developed,[48] the ongoing Add-Aspirin trial is investigating the effectiveness of regular aspirin for patients with non-metastatic colorectal, breast, gastro-oesophageal and prostate cancer.[49 50] Our findings suggest such groups may be particularly receptive to receiving a recommendation to use aspirin for preventive therapy.

Our study had a number of strengths. We used two well-established frameworks, the TDF and Necessity–Concerns Framework, to develop a comprehensive analysis of the barriers and motivators towards taking aspirin for colorectal cancer prevention among the general population. A particular strength of the TDF is that the framework can aid in specifying the beliefs and attitudes that are amenable to change[25] and offers an explicit framework for mapping onto behaviour change strategies (eg, *The Behaviour Change Wheel*).[51] The online survey also collected unique data on use of daily aspirin and reasons for taking the medication among the UK general population aged 50–70 years.

This study also had limitations. We recruited a moderate sample size of 400 respondents to the survey, therefore the findings should be generalised to the wider UK population with caution. In some cases, weighting approaches can be employed with the aim to weight participants' responses to represent the target population.[52] However, we concluded that weighting was inappropriate for our survey as several demographic (eg, ethnicity) categories contained little to no data, which can lead to over or under-representing responses for these population groups.[53] Self-selection to the survey and interviews may have also resulted in recruiting people with stronger views on the topic than those in the general population.[54]

For the interviews, most participants were white, with only three people recruited from an Asian background. Further research is warranted to explore the views of people from different ethnic minority groups on the use of aspirin for cancer preventive therapy. There are likely to be specific barriers among some ethnic minority groups. For example, research has observed South Asian respondents to view cancer as a taboo subject,[55 56] and have discussed the stigma attached to taking long-term medications,[57–59] which could reduce uptake of cancer preventive therapy.

## CONCLUSION

People with personal or family history of cancer were particularly receptive towards aspirin for colorectal cancer prevention. Future guidance recommending aspirin for cancer prevention may be of particular interest to these groups. Concerns about the side effects of aspirin were common across the sample, highlighting the need to support informed decisions on the medication. Guidance and advice recommending aspirin should be communicated from sources perceived as trustworthy by the public, such as healthcare professionals.

**Acknowledgements** The authors thank all the participants who took part in the study. We also acknowledge the AsCaP senior executive board and committee members.

**Collaborators** Aspirin for Cancer Prevention AsCaP Steering Committee Members: Professor Jack Cuzick, Queen Mary University of London. Chair. Professor Frances Balkwill, Queen Mary University of London. Professor Tim Bishop, University of Leeds. Professor Sir John Burn, Newcastle University. Professor Andrew T. Chan, Harvard School of Medicine. Dr Colin Crooks, University of Nottingham. Professor Chris Hawkey, University of Nottingham. Professor Ruth Langley, University College London. Ms Mairead McKenzie, Independent Cancer Patients' Voice. Dr Belinda Nedjai, Queen Mary University of London. Professor Paola Patrignani, Università "G. d'Annunzio" di Chieti-Pescara. Professor Carlo Patrono, Catholic University of the Sacred Heart, Rome. Dr Bianca Rocca, Catholic University of the Sacred Heart, Rome. Professor Samuel Smith, University of Leeds. Dr Laura Greaves, Newcastle University.

**Contributors** Conceptualisation: KEL, SGS, RF, LHH and LZ. Methodology: KEL, SGS, RF, LHH, LZ, DGT and MM. Supervision: SGS, RF, LHH and LZ. Funding acquisition: SGS, DGT and MM. Investigation: KEL. Formal analysis: KEL, SMCG and SGS. Writing—original draft: KEL. Writing—review and editing: all authors. Guarantor: KEL.

**Funding** This work is fully funded by the Aspirin for Cancer Prevention AsCaP Group CRUK Grant Code: A24991, Senior Executive Board Professor J Burn, Professor A T Chan, Professor J Cuzick, Dr B Nedjai, Professor Ruth Langley. During the study, KEL was supported by an Economic and Social Research Council studentship (grant number ES/P000745/1). KEL also acknowledges funding support from an ESRC Postdoctoral Fellowship (ES/Y00759X/1). This report is independent research supported by the National Institute for Health Research NIHR Advanced Fellowship, SGS, (grant number NIHR300588). SGS also acknowledges funding support from a Yorkshire Cancer Research University Academic Fellowship. The funders had no role in study design, data collection and analysis, decision to publish or preparation of the manuscript.

**Competing interests** RF is a member of NICE Implementation Strategy Group. All other authors have no competing interests to declare that are relevant to the content of this article.

**Patient and public involvement** Patients and/or the public were involved in the design, or conduct, or reporting, or dissemination plans of this research. Refer to the Methods section for further details.

**Patient consent for publication** Not applicable.

**Ethics approval** This study involves human participants. Ethical approval was awarded by the University of Leeds School of Medicine Research Ethics Committee (MREC 19-091). Verbal informed consent was obtained from all individual participants included in this study. Participants gave informed consent to participate in the study before taking part.

**Provenance and peer review** Not commissioned; externally peer reviewed.

**Data availability statement** Data are available in a public, open access repository. The survey data is available in a public, open access repository. The survey dataset and analysis scripts are publicly available in the Research Data Leeds Repository: https://doi.org/10.5518/1379. The qualitative data is available upon reasonable request. The methods and analysis plan were preregistered: https://doi.org/10.17605/OSF.IO/3EFG7.

**ORCID iDs**
Kelly E Lloyd http://orcid.org/0000-0002-0420-2342
Robbie Foy http://orcid.org/0000-0003-0605-7713
Samuel G Smith http://orcid.org/0000-0003-1983-4470

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
