## [Reviewer comments · BMJ Open]

ARTICLE DETAILS

TITLE (PROVISIONAL)	Acceptability of aspirin for cancer preventive therapy: a survey and qualitative study exploring the views of the UK general population
AUTHORS	Lloyd, Kelly; Hall, Louise; Ziegler, Lucy; Foy, Robbie; Green, Sophie M. C.; MacKenzie, Mairead; Taylor, David; Smith, Samuel

VERSION 1 – REVIEW

REVIEWER	Semedo, Lenira Cardiff University, Healthcare Sciences
REVIEW RETURNED	10-Sep-2023

GENERAL COMMENTS	Thank you for inviting review of your manuscript. The abstract section clearly describes the study aims and findings. The finding concerning side-effects could be clearly explained in the abstract (e.g. type and association with dose, harm)..Potentially refraining from using the term mix-methods as the analysis of the survey is purely descriptive. The qualitative data analysis is comprehensive and informed by a framework aligned to the aims of the study. The qualitative findings confirm previous exploratory work in the field. As you mention about motivators to take aspirin, it would have been interesting to know more about whether the public (survey responses) engaged in preventive health behaviours (cancer). I can see that this was asked in the interviews and I wonder whether this would be relevant to mention. The discussion is aligned to the findings and limitations are clearly stated. This paper reads and flows well. It confirms previous work undertaken. Slightly unsure of the novelty element.
--

REVIEWER	Archer, Stephanie Imperial College London, NIHR Imperial Patient Safety Translational Research Centre
REVIEW RETURNED	11-Sep-2023

GENERAL COMMENTS	Thank you for asking me to review this paper – it is generally well written and provides some useful insights into this topic area. I have noted a few suggestions below that will help to approve the manuscript. Abstract It feels a bit odd to talk about the Australian national guidance and its application to the UK general population in the abstract without any further explanation – I would consider removing this from here. Introduction
--

	In paragraph two, the authors may want to consider removing the reference to the lack of US guidance on the use of Aspirin. It doesn't seem to be relevant here. The point about people in the general population having already used Aspirin for something else doesn't really flow, and needs to be better integrated. At the end of the introduction it would have been better to state the aim of the study, rather than a brief synopsis of what was done (we find that out in the coming section). Methods In the design section, it would be helpful to swap round reasons 1 and 2 and then the bits about the interview can be better linked together, helping with the flow. I have an issue with saturation being used for reflexive thematic analysis without justification – I suggest the researchers read Braun & Clarke (2021) – DOI: 10.1080/2159676X.2019.1704846 and incorporate some additional text on its suitability here. In the data collection and analysis section, the authors report that a proportion of the data were double coded by authors – it isn't clear why this happened or what they authors were hoping to get from it – as with the above, this kind of practice doesn't really fit with reflexive TA (as it is very positivist) and needs further justifying. Going back to the new Braun & Clarke text (ISBN: 9781526417299) may be helpful here. Results The authors could consider changing the columns in Table 3 to have the barriers presented first – this would then match the order of the prose in the analysis (barriers are presented first). In theme 2, it seems odd to have the bit of text around Lynch and bleeding – this would fit better in the discussion than here. In the paragraph below this, there is an embedded quote and this disrupts the flow- it also appears to be the only one presented like this. It may help to pull this out and present it the same as the others. Discussion In the opening para, you talk about acceptability, but I'm not sure that you have actually addressed this (and you haven't used an theory of acceptability in your analysis). I suggest that you keep it more closely linked to what you have looked at in terms of barriers and facilitators. A bit more detail on how the TDF domains link with the NCF would be helpful – at the moment, this seems a bit superficial. The authors could consider moving the section on further work down to the end of the discussion to help tie up some of the suggestions from the limitations section (it would be nice to see some strengths listed too), and to help end this on a more positive note.
--	---

VERSION 1 – AUTHOR RESPONSE

Reviewer: 1

Dr. Lenira Semedo, Cardiff University

Comments to the Author:

Thank you for inviting review of your manuscript. The abstract section clearly describes the study aims and findings. The finding concerning side-effects could be clearly explained in the abstract (e.g. type and association with dose, harm).

Authors' reply: Thank you for your time and expertise in reviewing our manuscript. We have now expanded upon participants' concerns on aspirin side-effects (e.g., dose, type of harm) in the abstract:

"Concerns about taking aspirin at higher doses and its side-effects, such as gastrointestinal bleeding, were common." (Abstract, page 2).

Potentially refraining from using the term mix-methods as the analysis of the survey is purely descriptive.

Authors' reply: Thank you for this suggestion. We have now removed reference of mixed method from the study title and in the paper. The new title of the paper is "Acceptability of aspirin for cancer preventive therapy: a survey and qualitative study exploring the views of the UK general population" (Page 1).

The qualitative data analysis is comprehensive and informed by a framework aligned to the aims of the study. The qualitative findings confirm previous exploratory work in the field. As you mention about motivators to take aspirin, it would have been interesting to know more about whether the public (survey responses) engaged in preventive health behaviours (cancer). I can see that this was asked in the interviews and I wonder whether this would be relevant to mention.

Authors' reply: Thank you for this suggestion. We did not collect data in the survey on participants' other cancer preventive behaviours. In some interviews, other preventive cancer behaviours were briefly touched on. In most cases though, there were no salient findings relating these preventive behaviours to participants' barriers and facilitators to potentially taking aspirin for colorectal cancer prevention. Some participants felt that other options, such as diet, would be more effective for colorectal cancer prevention than taking medication, which we have included in the results. Another participant also felt that because they attended colorectal cancer screening that taking aspirin would not provide an additional benefit. We have now added this additional finding to the results:

"A few participants perceived other lifestyle changes, such as diet and attending preventive screening, as more effective and necessary than taking medication to prevent cancer.

"I think [colorectal cancer] can be broadly governed by diet, fibre, etc." (E.Y.).

"So I'm at the age now where I get a bowel screening, [...] So I'm having difficulty seeing what aspirin would add to the mix if I'm kind of okay at the moment." (R.C.)." (Results, page 12).

The discussion is aligned to the findings and limitations are clearly stated. This paper reads and flows well. It confirms previous work undertaken. Slightly unsure of the novelty element.

Authors' reply: Thank you for your positive comments on our paper. Whilst we are not the first to explore public attitudes to aspirin, few published studies have explored this, and our work adds to the growing body of evidence for aspirin as a promising but underused cancer prevention intervention. In addition, our theoretically-informed interviews, using the Theoretical Domains Framework, have enabled us to uniquely explore the modifiable beliefs and attitudes affecting use of aspirin.

Reviewer: 2

Dr. Stephanie Archer, Imperial College London

Comments to the Author:

Thank you for asking me to review this paper – it is generally well written and provides some useful insights into this topic area. I have noted a few suggestions below that will help to approve the manuscript.

Authors' reply: Thank you for your time and expertise in reviewing our manuscript.

Abstract

It feels a bit odd to talk about the Australian national guidance and its application to the UK general population in the abstract without any further explanation – I would consider removing this from here.

Authors' reply: We have now removed the reference to the Australian national guidance in the Abstract.

Introduction

In paragraph two, the authors may want to consider removing the reference to the lack of US guidance on the use of Aspirin. It doesn't seem to be relevant here.

Authors' reply: We have now removed the US national guidance from the Introduction.

The point about people in the general population having already used Aspirin for something else doesn't really flow, and needs to be better integrated.

Authors' reply: Thank you for this suggestion. We have now added a sentence at the end of this paragraph to integrate this point into the wider discussion on barriers and motivators towards aspirin for colorectal cancer prevention:

"Decisions on whether to use aspirin involve consideration of potential benefits and side-effects, such as an increased risk of gastrointestinal bleeding (9). For individuals at population risk of colorectal cancer, regular aspirin use between 75mg to 325mg appears to have a favourable benefit-harm profile (6), although the risk of side-effects increases substantially after age 70 (6, 10, 11). Current use and acceptability of aspirin for colorectal cancer prevention among the UK public is unknown (12). It is likely though that a proportion of the public have experience taking aspirin (13), such as for pain relief, or cardiovascular disease risk reduction (14). People's prior use of aspirin could support the implementation of the medication for the purpose of colorectal cancer prevention." (Introduction, page 4).

At the end of the introduction it would have been better to state the aim of the study, rather than a brief synopsis of what was done (we find that out in the coming section).

Authors' reply: Thank you for this suggestion, we have now added to discuss the aims of the study:

"In this study, we aimed to investigate aspirin use and associated knowledge among people from the UK general population. We also aimed to explore the potential facilitators and barriers towards taking aspirin for colorectal cancer prevention." (Introduction, page 5).

Methods

In the design section, it would be helpful to swap round reasons 1 and 2 and then the bits about the interview can be better linked together, helping with the flow.

Authors' reply: We have now swapped the order of reasons 1 and 2 for conducting the study:

"We conducted an online survey for two reasons: 1) to collect initial data on participants' prior use of aspirin and their knowledge on taking aspirin for cancer prevention; 2) to recruit people from the UK general population to an interview." (Methods, page 5).

I have an issue with saturation being used for reflexive thematic analysis without justification – I suggest the researchers read Braun & Clarke (2021) – DOI: 10.1080/2159676X.2019.1704846 and incorporate some additional text on its suitability here.

Authors' reply: Thank you for highlighting this. We acknowledge that data saturation is not the preferred approach of Braun and Clarke, but felt that the approach was useful in our study for ensuring we had a large enough sample for the analysis but did not waste resources by oversampling. The data saturation approach we used has also been recommended by Francis et al. (2010) for theory-based interview studies. In our methods section, we have now provided justification for our approach to use data saturation and the numerical criteria of 10 + 3, in line with the recommendation from Braun and Clarke 2021 paper:

“One author (KEL) recruited participants until data saturation was considered to have been achieved, following an established method for assessing data saturation (24). Data saturation was used to ensure that the sample size was large enough to provide informative data on the topic, but not so large as to waste resources and participants’ time. We used a 10 + 3 stopping criterion to assess for data saturation, which is a tested and recommended approach for theory-based interview studies (24). After 10 interviews, we assessed for data saturation and stopped recruitment once three subsequent interviews had been conducted and no new themes were identified (24). For example, recruitment would stop after participant 13, if no new themes were observed from participants 11, 12, and 13.” (Methods, page 5-6).

In the data collection and analysis section, the authors report that a proportion of the data were double coded by authors – it isn’t clear why this happened or what they authors were hoping to get from it – as with the above, this kind of practice doesn’t really fit with reflexive TA (as it is very positivist) and needs further justifying. Going back to the new Braun & Clarke text (ISBN: 9781526417299) may be helpful here.

Authors’ reply: Thank you for highlighting this. We did not use multiple coders to check for the positivism approach of assessing for reliability, but instead to collaboratively discuss other interesting perspectives on the data to enhance the interpretative depth of the findings, which is aligned with Braun and Clarke’s reflexive thematic analysis (e.g., Braun V, Clarke V. Conceptual and design thinking for thematic analysis. 2022). We agree that our rationale for using this approach is currently unclear in the manuscript, and have expanded upon this section:

“One author (KEL) coded and analysed all transcripts, and two authors (SGS, SMCG) double coded a proportion of interviews. The findings and potential themes were discussed collaboratively among the three authors, with the aim to explore new perspectives on the data and enhance interpretative depth (33).” (Methods, page 7).

Results

The authors could consider changing the columns in Table 3 to have the barriers presented first – this would then match the order of the prose in the analysis (barriers are presented first).

Authors’ reply: We have now swapped the order in the table with barriers column presented before facilitators (Page 11).

In theme 2, it seems odd to have the bit of text around Lynch and bleeding – this would fit better in the discussion than here.

Authors’ reply: Thank you for highlighting this. We have now amended this text:

“Many participants expressed concerns about the potential side-effects of daily aspirin use. Several participants expressed feeling more comfortable using a lower dose of daily aspirin (e.g., 75mg), due to the concerns about these side-effects.

“Because [150-300mg] seems to be a lot. [...], but 75mg, a small dose, I would feel comfortable with that.” (S.S.)” (Results, page 13).

In the paragraph below this, there is an embedded quote and this disrupts the flow- it also appears to be the only one presented like this. It may help to pull this out and present it the same as the others.

Authors’ reply: Thank you for this suggestion. We have now displayed this quote outside of the text, similar to how the other quotes are displayed in the paper:

“Several participants’ concerns regarding aspirin appeared to be related to the perceived harm of taking any long-term medication, and the perception that taking daily medication would mean that they were an ‘unhealthy’ person.

“[From taking medication] you suddenly step from being a healthy person to a potentially unhealthy person because it’s a medical intervention” (R.C.)” (Results, page 13).

Discussion

In the opening para, you talk about acceptability, but I’m not sure that you have actually addressed this (and you haven’t used an theory of acceptability in your analysis). I suggest that you keep it more closely linked to what you have looked at in terms of barriers and facilitators.

Authors’ reply: We have now changed the opening paragraph of the Discussion to focus on the barriers and facilitators to taking aspirin:

“This study of the UK general population found low reported current use of aspirin for cancer prevention among survey respondents, and several barriers and motivators towards taking aspirin for this purpose across interviewees.” (Discussion, page 15).

A bit more detail on how the TDF domains link with the NCF would be helpful – at the moment, this seems a bit superficial.

Authors’ reply: Thank you for this suggestion, we have now added further detail on how the TDF findings relate to the Necessity-Concerns Framework:

“Participant beliefs about the consequences of taking aspirin, such as the necessity for and side-effects of the medication, were particularly important, and relate to the Necessity-Concerns Framework (30). Several participants expressed specific beliefs and concerns on taking aspirin for cancer prevention, as well as general concerns on taking any medication daily.” (Discussion, page 15).

The authors could consider moving the section on further work down to the end of the discussion to help tie up some of the suggestions from the limitations section (it would be nice to see some strengths listed too), and to help end this on a more positive note.

Authors’ reply: Thank you for this suggestion. As many of our suggestions for future research are intertwined with discussions on previous research in this area, we have decided to keep the current organisation of the Discussion with limitations presented at the end. We have though included your suggestion to report the strengths of the paper with limitations:

“Our study had a number of strengths. We utilised two well-established frameworks, the TDF and Necessity-Concerns Framework, to develop a comprehensive analysis of the barriers and motivators towards taking aspirin for colorectal cancer prevention among the general population. A particular strength of the TDF is that the framework can aid in specifying the beliefs and attitudes that are amenable to change (25), and offers an explicit framework for mapping onto behaviour change strategies (e.g., the Behaviour Change Wheel) (52). The online survey also collected unique data on use of daily aspirin and reasons for taking the medication among the UK general population aged 50 to 70.” (Discussion, page 16).

VERSION 2 – REVIEW

REVIEWER	Archer, Stephanie Imperial College London, NIHR Imperial Patient Safety Translational Research Centre
REVIEW RETURNED	20-Nov-2023
GENERAL COMMENTS	Thank you for taking the time to revise this manuscript to such a high standard. You have made an excellent job of incorporating our suggestions/feedback and your response letter was really easy to follow, making our job as a reviewer much easier!